# Act-to-Ground: A Framework for Symbol Grounding in Planning Domains

**Panagiotis Lymperopoulos**                                     PLYMPE01@TUFTS.EDU
and **Liping Liu**                                               LIPING.LIU@TUFTS.EDU
*Tufts University, Medford, MA, USA*

**Editors:** Leilani H. Gilpin, Eleonora Giunchiglia, Pascal Hitzler, and Emile van Krieken

Neurosymbolic decision-making agents inherit many of the critical transparency and interpretability benefits of planning-based symbolic agents but also face one of their central challenges: the Symbol Grounding Problem (SGP). Grounding hand-crafted symbolic planning domains to percepts typically requires training models with extensive annotated data which hinders their applicability to broader problems. In this work we propose Act-to-Ground (A2G), a framework for training grounding models for symbolic planners with weak supervision obtained through environment interaction or demonstrations. We first cast the grounding problem as an inference problem and 1) use satisfiability-based planning to provide weak supervision to the grounding model by exploiting knowledge already built into the planning domain, 2) propose an MCMC sampler that enables sampling weak labels for grounding planners, 3) improve neurosymbolic grounding performance via a score-matching objective and 4) propose a learnability condition for learning grounding models for planners.

## 1. Introduction

Neurosymbolic architectures that combine neural models with symbolic reasoning face a difficult challenge already identified in cognitive modeling: the symbol grounding problem (SGP) (Harnad, 1990; Wang et al., 2019; Yu et al., 2023). The SGP is the problem of connecting the abstract symbols internal to reasoning systems such as symbolic planners to subsymbolic percepts originating from outside the system. While this is a challenging problem, the benefits of transparency and interpretability enjoyed by symbolic systems, in conjunction with the successes of deep-learning models in computer-vision and other domains have, in recent years, motivated efforts towards addressing the SGP by learning deep models to ground symbols on perceptions. This allows for the construction of hybrid systems with perception and reasoning components that can be independently evaluated, verified and modified if necessary.

An important area of application for symbol grounding models is in grounding symbolic decision-making systems such as planners. Grounding these systems to sub-symbolic observations can greatly expand the applicability of symbolic planners, but training grounding models for planners is hard due to the need of labeled examples of environment states (Lv et al., 2022; Huang et al., 2019). To mitigate the data scarcity, many approaches forgo supervision altogether and learn to plan in expressive latent-spaces in an unsupervised manner Asai (2019); Asai and Fukunaga (2018); Xu et al. (2019); Barbin et al. (2022); Hafner et al. (2022); Umili et al. (2021). While they can learn to plan effectively, these methods typically

lose interpretability and transparency in decision making as planning no longer takes place in interpretable human-designed planning domains.

In practical applications such as robotics, grounding planners to sub-symbolic perceptions is crucial to enable high-level task planning. In those applications, the SGP is often tackled by a combination of pre-trained models for related tasks such as object detection Martinez-Martin and Del Pobil (2020); Mallick et al. (2018) and manually constructed heuristics which can work in controlled environments but require extensive engineering efforts to construct. A related well-studied problem is grounding natural language utterances and resolving object references using images and 3D point clouds Prabhudesai et al. (2019); Achlioptas et al. (2020); Hsu et al. (2023); Hong et al. (2023). The problem is distinct from grounding planning systems however, as human-specified planning domains are not necessarily expressed in natural language and can involve symbols that cannot be determined by object detection alone.

In this work, we propose Act-to-Ground (A2G), a framework for training grounding models for a symbolic planners without labeled data by exploiting the domain knowledge already encoded within the planning domain in conjunction with weak supervision signals obtained through environment interactions or observed trajectories. In summary, our contributions are: 1) we use satisfiability-based planning (Kautz et al., 1992; Kautz and Selman, 2006; Gocht and Balyo, 2017) to enable learning grounding models for symbolic planners, 2) we propose an MCMC sampler with randomized projections to enable sampling weak labels for grounding planning problems, 3) we improve grounding model performance via a score-matching objective Meng et al. (2022) and 4) propose a learnability condition for neurosymbolic symbol grounding of planning domains related to the learnability of weakly supervised learning. We evaluate our approach on two proposed learning settings for these neurosymbolic agents: learning by interaction and learning from demonstrations. We also independently evaluate the efficacy of our score-matching objective by comparing to previous work on neurosymbolic learning.

## 2. Related Work

The SGP poses a major challenge for neurosymbolic systems, particularly because obtaining annotated data to train models with full supervision can be prohibitively expensive. Recent work on end-to-end neurosymbolic learning (Daniele et al., 2022; Wang et al., 2019; Garnelo et al., 2016) where the symbolic reasoning system is learned directly from data also faces the SGP (Chang et al., 2020; Topan et al., 2021). While these methods achieve impressive results in learning hybrid neural and symbolic functions, it is difficult to interpret and verify the correctness of components separately. In this work, we want to verify the correctness of the perception component so that later changes to the symbolic components that do not affect perception can be made without hurting the agent's performance.

Focusing on planning problems, some systems learn discrete latent representations from observations of state transitions and automatically construct planning domains in that latent space (Asai, 2019; Asai and Fukunaga, 2018; Barbin et al., 2022; Umili et al., 2021), eliminating the need to annotate observations with symbolic states. Related methods (Mao et al., 2022; Umili et al., 2024) also focus on latent-space planning by learning structured transition models jointly with symbol grounding from trajectories. While the latent-spaces

learned can be effective for planning or reasoning, they can be difficult to interpret and make decision-making difficult to modify after training. Other methods adopt fully neural architectures for planning in latent space (Hafner et al., 2022; Xu et al., 2019). These methods circumvent the SGP by only operating on neural representations. While they show impressive planning and generalization abilities, they also tend to lack sufficient transparency and interpretability and it is not possible to separate the perception from the decision making sub-systems.

One approach for learning grounding models using information from planning domains uses observations of state transitions annotated with the actions that induced them (Migimatsu and Bohg, 2022) and derive weak labels for the grounding model using the action preconditions and effects. While related to our work, we consider the more general problem of deriving weak supervision from entire plans derived either from interacting with the environment or given as observations. Another line of work (Li et al., 2024; Huang et al., 2021) learns grounding models for human-designed reasoning systems but they are not directly applicable to planning problems. Also related are works on learning models with constraints (Xu et al., 2018; Ahmed et al., 2022, 2023) but those methods typically assume a fixed constraint set. By decoupling the constraint set from the neural model itself we allow full flexibility to learn accurate grounding models and also allow for the trained model to be used by different agents that rely on the same grounding.

Another approach is to use large-scale pretrained multi-modal models (Zhang et al., 2023) to perform the grounding. These models, trained on natural image-caption pairs show remarkable ability in providing natural language descriptions of natural scenes but are typically not trained to produce fine-grained descriptions and thus may suffer in some domains.

## 3. Method

In this section, we introduce A2G, a framework for learning grounding models for neurosymbolic planning agents. We first describe how weak labels for the grounding model are obtained via satisfiability-based planning and discuss two learning settings. We then formulate the grounding problem as statistical inference, relate it to weakly-supervised learning, and introduce a novel MCMC sampler for weak label sampling along with a score-matching objective used in A2G.

**Neurosymbolic Grounding via SAT-based Planning**  We define a classical planning domain as $\Sigma = (\mathcal{S}, \mathcal{O}, L)$, where $\mathcal{S}$ is the set of states, $\mathcal{O}$ the set of operators, and $L$ the set of fluents. A state $s \in \mathcal{S}$ assigns values to all fluents $l \in L$, and planning operators are specified by preconditions and effects (Russell and Norvig, 2016). In a neurosymbolic planning task, instead of directly observing the initial state $s_0$, the agent receives a low-level representation $\mathbf{x} \sim p(\mathbf{x} \mid s_0)$ (e.g., an image) and a goal description $s_g$. The agent must generate a plan $P$ to reach a goal state from $s_0$.

The goal is to recover $s_0$ from $\mathbf{x}$ using a learning model, where the initial state is represented by a vectorized label assignment $\mathbf{z} \in \mathcal{Z}$. The model is denoted as $p_\theta(\mathbf{z} \mid \mathbf{x})$. The space $\mathcal{Z}$ is discrete, structured by the planning problem. For instance, in a maze environment, $\mathbf{z}$ may encode attributes like agent location or wall positions, with entries $\mathbf{z}_i$ indicating specific states (e.g., $\mathbf{z}_i = a$ means the agent is at cell $i$).

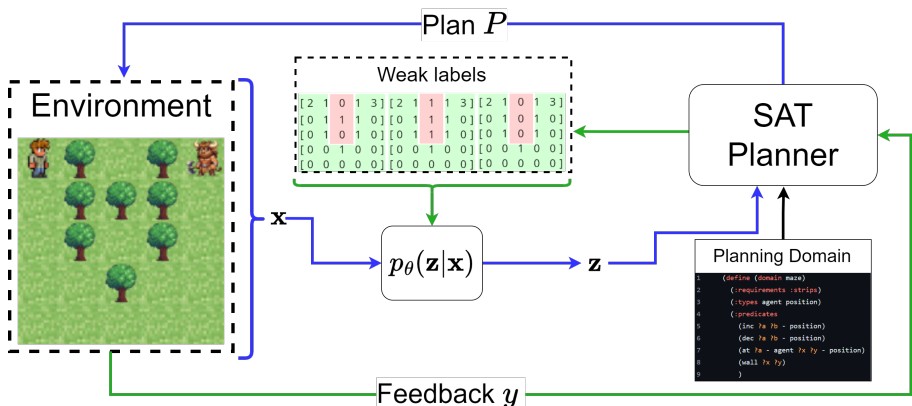

Figure 1: Overview of the learning by interaction setting for A2G. Blue arrows indicate the forward pass where the agent predicts a grounding and derives a plan to execute in the environment from it. The green arrows indicate the backward pass, where feedback from the environment is translated into constraints for the satisfiability-based planner, which in turn provides weak labels for the grounding model. The maze in the example however contains an area not reachable by any plans (red), making it impossible to distinguish between some groundings.

A SAT-encoding of a planning task is a formula $F = \text{encode}(\Sigma, s_g)$, where any satisfying assignment to variables in $F$ corresponds to a valid plan. A subset of variables in $F$ represents $s_0$, which must be inferred from input and environment interactions. The grounding model predicts $s_0$ to determine these variables and generate a plan $P$. The agent receives feedback $y$, which must be expressible as a constraint in $F$, such as verifying that executing $P$ led to the goal. Given $F$ and $y$, we define a constraint set $\mathcal{C}(y, P, F) \subseteq \mathcal{S}$ such that all $s_0 \in \mathcal{C}$ are consistent with feedback $y$ and planning constraints. Similarly, $\mathcal{C}$ constrains labels $\mathbf{z}$ of raw input $\mathbf{x}$. With a slight abuse of notation, we let $\mathcal{C}$ denote this constraint set over $\mathbf{z}$. Before discussing model training, we introduce two key methods for forming these constraint sets.

**Learning by Interaction:** In this learning setting the neurosymbolic agent interacts with the environment to obtain supervision for its grounding model. Upon receiving the observation $\mathbf{x}_0$, the agent samples a $\mathbf{z} \sim p_\theta(\mathbf{z}|\mathbf{x}_0)$ from the model and attempts to find a plan $P$. Otherwise it selects a random plan. It then executes the plan, observes $y$ and forms the set $\mathcal{C}$. In different problems the environment feedback $y$ can take different forms, such as whether the plan $P$ was successful. Future work can consider more sophisticated planning strategies that can use information from the grounding model or the structure of the planning domain to obtain informative supervision. Figure 1 summarizes the information flow in our architecture in this setting.

**Learning From Demonstrations:** In this learning setting, the neurosymbolic agent observes a fixed dataset of trajectories $D = \{(\mathbf{x}^i, P^i, y^i), i = 1 \ldots n\}$. In other words, the agent observes a fixed dataset of initial observations, plans and feedback for those plans. In general, both successful and unsuccessful trajectories can be included in the dataset, as they

can both be informative. As before, the agent can then form the set $\mathcal{C}^i$ for each observed $(\mathbf{x}^i, P^i, y^i)$.

There are other ways to form constraint sets depending on the available supervision. For instance in the special case where 1 step demonstrations are observed, the learning setting is similar to Migimatsu and Bohg (2022). In the rest of the work we focus on the two learning settings above.

### 3.1. Learning the grounding model

The problem of learning $p_\theta(\mathbf{z}|\mathbf{x})$ can be considered with weakly supervised learning since we have partial information about the label $\mathbf{z}$ from its constraints. The problem has been investigated by (Li et al., 2024). Here we give it a formal treatment from the perspective of statistical inference. The model should maximize the probability of $\mathbf{z} \in \mathcal{C}$.

$$\max_\theta \mathcal{L}(\theta) = \log \sum_{\mathbf{z} \in \mathcal{C}} p_\theta(\mathbf{z}|\mathbf{x}) \tag{1}$$

The problem is nontrivial for general problems as it requires the enumeration of $\mathbf{z}$ from the constraint set $\mathcal{C}$. To overcome the difficulty, one method is to find a lower bound of the log-likelihood. In the spirit of variational inference (Blei et al., 2017), we introduce another distribution $q(\mathbf{z}|\mathbf{x}, \mathcal{C})$ and then get a lower bound of the original objective

$$\mathbb{E}_q\left[\log p_\theta(\mathbf{z}|\mathbf{x}) - \log q(\mathbf{z}|\mathbf{x}, \mathcal{C})\right] \leq \mathcal{L}(\theta), \tag{2}$$

which is the negative KL divergence from $q$ to $p_\theta$. The bound is tight when

$$q(\mathbf{z}|\mathbf{x}, \mathcal{C}) = \begin{cases} \frac{p_\theta(\mathbf{z}|\mathbf{x})}{\sum_{\mathbf{z} \in \mathcal{C}} p_\theta(\mathbf{z}|\mathbf{x})} & \text{if } \mathbf{z} \in \mathcal{C} \\ 0 & \text{o.w.} \end{cases} \tag{3}$$

We reach the same training objective as (Li et al., 2024) but from a different perspective. Note that it is typically difficult to sample from the distribution $q$ when $p_\theta$ is parameterized by a neural network. We later discuss how we employ MCMC sampling from $q$ to optimize the objective. This objective is well connected to the theory of weakly supervised learning. As long as the distribution $p_\theta(\mathbf{z}|\mathbf{x})$ places zero probability outside of $\mathcal{C}$ for a given $\mathbf{x}$, it maximizes the objective in (1), which means the problem might be underdetermined. Therefore, our hope is that multiple training instances provided by all $(\mathcal{C}, \mathbf{x})$ pairs could provide sufficient information for learning the grounding model. We summarize the analysis in the following theorem.

**Learnability of neurosymbolic symbol grounding:** Here we informally state a learnability condition for the grounding model. A formal treatment in relation to the learnability of weakly supervised learning is available in the appendix.

**Theorem 1** *(Informally) Neurosymbolic grounding for planning agents is learnable if for a given initial state $s_0 \in S$, the corresponding observation $\mathbf{x} \in \mathcal{X}$ and two possible groundings, an incorrect $\tilde{\mathbf{z}} \in \mathcal{Z}$ and the correct $\mathbf{z}^* \in \mathcal{Z}$, an agent observes at least one piece of supervision $y$ (e.g. a successful plan) that is consistent with one grounding but not the other.*

In terms of the learning by interaction setting, if there are aspects of an environment state $s_0$ that are never involved in any plan the agent may execute, then it is not possible for the agent to determine the correct grounding for $s_0$. In terms of the learning from demonstrations setting, the source distribution of trajectories additionally needs to place some probability to sampling such plans and feedback. Figure 1 displays one such case, where it is not possible for the agent to form a plan that differentiates the inaccessible parts of the maze.

## 3.2. Improving Training of Neurosymbolic Grounding Models via Score Matching

Minimizing the KL-divergence in (2) trains grounding models but has limitations. If $p_\theta$ assigns mass outside $\mathcal{C}$, the gradient becomes uninformative. This issue arises because optimizing (2) w.r.t. $\theta$ tends to make $p_\theta$ cover $q$, a well-known effect. The problem worsens in high-dimensional $\mathcal{Z}$ due to the curse of dimensionality. To address this, we introduce a score-matching (SM) objective (Meng et al., 2022) that encourages $p_\theta$ to stay within $q$'s support, improving approximation and stabilizing training while efficiently reusing MCMC samples. The score matching objective between the model $p_\theta$ and the distribution $q$ is:

$$
\begin{aligned}
J_{sm}(\mathbf{x}, y) &= \mathbb{E}_{\mathbf{z}', \mathbf{z} \sim r(\mathbf{z}|\mathbf{x})}[d(\mathbf{z}, \mathbf{z}', \mathbf{x}, \mathcal{C})] \\
d(\mathbf{z}, \mathbf{z}', \mathbf{x}, \mathcal{C}) &= (\log \frac{p_\theta(\mathbf{z}'|\mathbf{x})}{p_\theta(\mathbf{z}|\mathbf{x})} - \log \frac{q(\mathbf{z}'|\mathbf{x}, \mathcal{C})}{q(\mathbf{z}|\mathbf{x}, \mathcal{C})})^2.
\end{aligned}
\tag{4}
$$

We choose to use $p_\theta(\mathbf{z}|\mathbf{x})$ as the sampling distribution $r$, but we don't need to consider the gradient of $\theta$ for sampling. Given the form of the optimal $q$ distribution in (3), if $\mathbf{z}'$ and $\mathbf{z}$ are both in $\mathcal{C}$, then the objective is zero. If they both are outside of $\mathcal{C}$, then $q$ has zero probabilities for both of them, providing no information about the ratio in $p_\theta$. We only need to consider the case when one of them is in $\mathcal{C}$ and the other is not. Let $\mathbf{z}' \in \mathcal{C}$ and $\mathbf{z} \in S \backslash \mathcal{C}$. While technically $q(\mathbf{z}|\mathbf{x}, \mathcal{C})$ is zero in this case, we encourage the model to have a much larger probability at $\mathbf{z}'$ than $\mathbf{z}$. For example, the difference of log probability is a large number $\omega \in \mathbb{R}$, say 1000. Then, we compute the loss:

$$
l(\mathbf{z}', \mathbf{z}, \mathbf{x}; \theta) = \begin{cases} -\log \frac{p_\theta(\mathbf{z}'|\mathbf{x})}{p_\theta(\mathbf{z}|\mathbf{x})}, & \log \frac{p_\theta(\mathbf{z}'|\mathbf{x})}{p_\theta(\mathbf{z}|\mathbf{x})} \leq \omega \\ 0, & \text{otherwise}, \end{cases}
\tag{5}
$$

and form the proxy objective:

$$
\hat{J}_{sm}(\theta) = \mathbb{E}_{\mathbf{z}', \mathbf{z} \sim r(\mathbf{z}|\mathbf{x})} \left[ \mathbf{1}(\mathbf{z}' \in \mathcal{C} \wedge \mathbf{z} \notin \mathcal{C}) l(\mathbf{z}', \mathbf{z}, \mathbf{x}; \theta) \right].
\tag{6}
$$

The gradient for (6) w.r.t. $\theta$ is easy to compute using automatic differentiation. To get a sample $\mathbf{z}'$ we can re-use samples from the $q$ distribution we obtain through MCMC sampling, which will be discussed later. The sample $\mathbf{z}$ is obtained from the model $p_\theta(\mathbf{z}|\mathbf{x})$. Initially, $\mathbf{z}$ is very likely to not be in $\mathcal{C}$; when it is hard to draw a $\mathbf{z} \notin \mathcal{C}$, it means $p_\theta$ is well trained. In the calculation, we check $\mathbf{z}$ against the constraint set $\mathcal{C}$ which is not very computationally expensive in practice by making use of incremental SAT-solving (Gocht and Balyo, 2017) to accelerate the computation.

We combine the new score-matching training objective with the KL-divergence objective. Note that both objectives force $p_\theta$ move toward the same optimal distribution (3).

$$\min_\theta E_q[-\log p_\theta(\mathbf{z}|\mathbf{x})] + \alpha \hat{J}_{sm}(\theta) \tag{7}$$

where $\alpha$ is a hyperparameter controlling the balance between the two terms. Here the expectation with respect $q$ is estimated with a few MCMC samples from $q$. In the optimization procedure, we often apply the annealing technique over $q$ to improve the learning performance: $q(\mathbf{z}|\mathbf{x}, \mathcal{C}) \propto (p_\theta(\mathbf{z}|\mathbf{x}))^{\frac{1}{\gamma}}$ for $\mathbf{z} \in \mathcal{C}$. Here $\gamma$ can be annealed during training.

By improving the approximation of $q$ by $p_\theta$, the auxiliary objective also helps improve training consistency. This is because the gradient of $\hat{J}_{SM}$ is specifically designed to be informative even if $p_\theta$ places low probability mass inside $\mathcal{C}$. This is especially important when the space of groundings $\mathcal{Z}$ is high dimensional, as in that setting it is more likely for $p_\theta$ to place low probability in $\mathcal{C}$ due to the curse of dimensionality.

### 3.3. MCMC sampler for sampling from $q$ in planning problems

Since it is not possible to sample from $q$ directly and rejection sampling is difficult to apply when $\mathcal{Z}$ has high dimensions, we use MCMC sampling. First, we initialize a chain by sampling $\mathbf{z} \sim p_\theta(\mathbf{z}|\mathbf{x})$ and projecting the sample inside of $\mathcal{C}$. The projection is computed by randomly masking entries of $\mathbf{z}$ and filling them in with a SAT solver, increasing the masking probability until a solution is found. The proposal distribution acts similarly: We first perturb the previous $\mathbf{z}$ by uniformly changing some of its entries and then we project the perturbed grounding back into $\mathcal{C}$. Then the acceptance ratio is computed using $q$, which in turn depends on $p_\theta$. By masking randomly and gradually increasing the masking probability, we ensure that we always reach another element of $\mathcal{C}$. Algorithm 1 in the appendix formally describes the procedure.

## 4. Experiments

In this section we present three experiments to validate our methodology. In our first experiment, we demonstrate the learning by interaction paradigm for grounding neurosymbolic agents in a maze navigation domain. In the second experiment, we demonstrate the learning by demonstration paradigm using photorealistic renderings of Blocksworld, a traditional planning problem. In the final experiment we demonstrate the improved grounding performance of our method by comparing against previous work in grounding hand-written equations.

| Method | Grounding Accuracy |
|--------|--------------------|
| SoftenSG | $59.49 \pm 4.23$ |
| A2G | $\mathbf{65.32 \pm 2.83}$ |
| Supervized | $86.46 \pm 0.28$ |
| GPT-4o | 53.33 |
| Llama3.2-11b | 44.18 |
| LlaVA-13b | 35.80 |

Table 1: Comparison of methods in learning by interaction in the maze domain.

**Evaluation metrics:** Since we focus on training grounding models for planners, we evaluate methods based on their accuracy in grounding individual symbols over the test set. However, in the handwritten-formula experiment we also include the accuracy of the final computation result for consistency with previous

work. The main baseline we compare against is SoftenSG (Li et al., 2024). In experiments 1 and 2 we also include include an unrealistic fully supervised baseline, where a model is trained with ground truth groundings to obtain a theoretical limit of performance for weakly supervised methods and 3 multimodal models GPT-4o, Llama3.2-vision-11b, and LLaVA-13b (Liu et al., 2023). Additional details on prompts and model settings are available in the Appendix.

### 4.1. Experiment 1: Learning by Interaction

In this experiment we evaluate our approach to learning grounding models in the learning by interaction setting in a maze navigation domain.

**Task:** We randomly generate mazes of size $5 \times 5$ and uniformly select two free positions as the starting and goal position. We render each maze into a single image. The space of groundings $\mathcal{Z}$ consists of 25 categorical variables of 4 categories each, corresponding to the states of cells in the maze, namely agent, goal, wall or clear. The environment feedback $y$ indicates whether each step of the plan achieved a goal state. We generate 800 mazes for training, 100 for validation and 100 for testing. The code used to generate the dataset and the SAT encoding of the maze problem are available alongside our training code.

**Experiment Settings:** We run a maximum of 500 training epochs and use early stopping with patience 50 based on the grounding accuracy over the validation set. In each training epoch we sample batches of 64 mazes, run an episode in each one in parallel and update the model. As a result we run a maximum of 500 episodes per maze. We omit SSL from this comparison as its performance is not competitive. We also run a fully supervised benchmark for comparison. We use a ResNet pre-trained on the ImageNet-1000 dataset as the backbone of the model and replace the prediction head with a simple MLP. Finally, we use our modified MCMC sampler for both methods. Additional reproducibility details are available in the appendix.

**Results:** Table 1 summarizes the results of this experiment. Due to the large output space, the task is very difficult for both training methods. In fact, even the fully supervised benchmark cannot perfectly learn the task. Nevertheless, A2G offers significant performance improvement over SoftenSG. The large output space means that the KL term alone may not have sufficiently informative gradients for many samples since it leads to approximations that cover $q$, making training inefficient. In addition, since the SM loss in A2G strongly encourages the model to predict valid groundings, part of the performance improvement may also be due to additional supervision through environment interaction.This is evident from the increase in samples usable for plan construction during A2G training, which rises to 5-10%, compared to less than 1% with the baseline. This aligns with the previously discussed learnability condition, where more diverse supervision helps resolve ambiguities and improve grounding accuracy. Task success results are provided in the appendix for completeness. Interestingly. the pretrained foundation models perform relatively poorly as they are not trained for producing fine-grained descriptions

### 4.2. Experiment 2: Learning by Demonstration

In this experiment we evaluate A2G in the learning from demonstrations setting with trajectories generated using Photorealistic Blocksworld (Asai, 2018).

**Dataset:** Blocksworld is a classical planning problem where objects are either stacked or placed on a table. We generate problems with 5 objects by creating a random state, then executing random valid actions to reach a goal state. The optimal plan between these states serves as the demonstration, with $y$ consisting of the plan and goal state. To enhance the supervision signal, we include the goal state description when constructing constraints. A grounding $\mathbf{z}$ includes 5 categorical variables, each with 6 categories representing possible object positions (the table or other objects). We create two datasets: *Blocksworld* with 1000 samples and *BlocksworldLarge* with 3000 samples, using an 80/10/10 split for training, validation, and testing. The SAT encoding is available with our training code.

| Method | Grounding Accuracy | |
| --- | --- | --- |
| | **BlocksWorld** | **BlocksWorldLarge** |
| SoftenSG | $31.37 \pm 7.43$ | $41.87 \pm 6.31$ |
| A2G | $\mathbf{73.52 \pm 3.63}$ | $\mathbf{91.15 \pm 4.23}$ |
| Supervized | $97.93 \pm 0.47$ | $100 \pm 0.00$ |
| GPT-4o | 43.30 | - |
| Llama3.2-11b | 10.33 | - |
| LlaVA | 6.33 | - |

Table 2: Comparison of methods in the learning by demonstration paradigm in the BlocksWorld domain.

**Experiment Settings:** We run a maximum of 500 training epochs and use early stopping with patience 50 based on the grounding accuracy over the validation set. As before, we omit SSL from this comparison as its performance is not competitive and only run *Stage 1* training. We also run a fully supervised benchmark for comparison. We use a ResNet pre-trained on ImageNet-1k as the backbone of the model and replace the prediction head with 5 different MLPs, one for each object in the scene. Finally, we use our modified MCMC sampler for both methods. Additional reproducibility details are available in the appendix

**Results:** As shown in Table 2, A2G significantly outperforms the baseline, likely due to better approximation of the $q$ distribution. However, it still lags behind full supervision, though it nearly matches it on the larger dataset. This gap may stem from the learnability challenge identified in Section 3: in Blocksworld, optimal plans can solve multiple initial states, potentially omitting details needed for accurate grounding. The larger dataset mitigates these ambiguities by providing more varied plans, suggesting that demonstration datasets should include diverse, exploratory trajectories to better constrain symbol grounding. Task success results are in the appendix. Pretrained foundation models remain weak, with A2G significantly outperforming them, as these models generate high-level descriptions (e.g., "the red cylinder is on the left") rather than fine-grained ones.

### 4.3. Experiment 3: Improving Training Consistency

In this experiment, we evaluate the learning performance of our method on a simpler problem to validate the efficacy of our score-matching objective. We compare our algorithm against SoftenSG (Li et al., 2024), as well as SSL, a stochastic variant of the semantic loss Xu et al. (2018) as the original is not tractable in this setting, on a handwritten for-

mula dataset (Li et al., 2020). While A2G enables learning grounding models in planning domains, we also evaluate in this setting to have a side-to-side comparison with prior work.

**Dataset:** Following Li et al. (2024), we limit formulas to length 7, using 20% of training data for validation and the test set for testing. Each formula is a sequence of images, classified into 14 classes (digits 0-9 and operators $+, -, \times, \div$). Supervision is a numerical result of the formula's computation.

**Experiment Settings:** We compare only with the top two methods from (Li et al., 2024) due to the large gap with others: SSL Xu et al. (2018) and the approach by Li et al. (2024). A simple convolutional network is used for all methods. Models train for up to 2000 epochs with early stopping after 200 epochs of no validation improvement. We perform training in two stages with different settings of $\gamma$. Additional details are available in the Appendix.

| Method | Grounding Accuracy | Result Accuracy |
|---|---|---|
| SSL | $67.01 \pm 2.62$ | $6.29 \pm 1.49$ |
| SoftenSG | $57.24 \pm 41.35$ | $44.55 \pm 43.89$ |
| A2G | $\mathbf{98.59 \pm 0.16}$ | $\mathbf{90.42 \pm 1.05}$ |

Table 3: Comparison of methods on the handwritten formula dataset, averaging over 8 runs for each method and reporting average and standard deviation grounding and result accuracy. The addition of the score-matching objective stabilizes training and results in better performance on average.

**Results:** Table 3 summarizes the results. Our SoftenSG results at first glance deviate from the original paper, but the best run achieves 98.9% grounding accuracy and 92.6% result accuracy, closely matching the values reported in the original paper, and A2G's average. However, averaging over multiple runs reveals high variance and lower mean performance due to training instability.

Figure 2 (Appendix) illustrates this instability: some runs collapse, predicting the same class for all inputs, likely due to low probability mass within constraints, rendering gradients uninformative. Some runs recover quickly, while others do not. Consequently, SoftenSG has lower mean performance and higher variance than A2G. The SM objective in A2G ensures informative gradients even when valid solutions have low probability, stabilizing training.

## 5. Discussion

In this work we propose A2G, a framework for training neural grounding models for neurosymbolic planning agents with weak supervision. While we focus on two ways to obtain supervision, our framework is general and compatible with other sources of knowledge gathered from the environment or obtained from an expert in a human-in-the-loop setting. We leave the exploration of such supervision schemes to future work.

One limitation of A2G when learning by interaction is that it does not incorporate sequential information from the trajectory. In the environments we consider that is not a major problem, however, in partially observable domains this may not suffice. Future work can explore how to interleave sensing and decision making to maximize symbol grounding supervision. In turn this can inform trajectory collection for training from demonstration. In addition, our learnability condition indicates that weak supervision may not be sufficient for training accurate grounding models in all domains and points to using unsupervised methods such as contrastive learning to supplement model training.

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

## Appendix

### Learnability of neurosymbolic symbol grounding

The framework of neurosymbolic symbol grounding corresponds to a weakly-supervised learning problem, where each observed $y$ defines the set of labels $\mathcal{C}$. Consider a tuple $(\mathbf{x}, y)$ sampled from the data distribution $p(\mathbf{x}, \mathbf{y})$. By construction, the true grounding $\mathbf{z}^* \in \mathcal{Z}$ is always in the label set $\mathcal{C}$ and other $\mathbf{z} \in \mathcal{C}$ are *distractor labels*. We can consider the *ambiguity degree* (Cour et al., 2011) for this problem:

$$\gamma = \sup_{\substack{(\mathbf{x},y) \in \mathcal{X} \times \mathcal{Y}, \mathbf{z} \in \mathcal{Z}: \\ p(\mathbf{x},y) > 0, \mathbf{z} \neq \mathbf{z}^*}} Pr_{(\mathbf{x},y) \sim p(\mathbf{x},y)}(\mathbf{z} \in \mathcal{C}). \tag{8}$$

The problem is *learnable* in the *small ambiguity degree condition*, where $\gamma < 1$ (Liu and Dietterich, 2014). Simply put, as long as a distractor label does not always co-occur with the true label, then the problem is learnable. We now connect this learnability condition with planning problems to establish the learnability of neurosymbolic grounding for planning agents.

Consider the neurosymbolic grounding problem, where $\mathbf{x} \in \mathcal{X}$ is a representation of $s_0$ (e.g. an image), $\mathbf{z} \in \mathcal{Z}$ describes $s_0$ and $y$ is the observed feedback. These define a set $\mathcal{C}$ as before. It is easy to see that if there is an initial state $s_0$ represented by $\mathbf{x}$ and with correct grounding $\mathbf{z}^*$ such that:

$$\forall y \in \mathcal{Y}, \ p(\mathbf{x}, y) > 0, \ \exists \tilde{\mathbf{z}} \in \mathcal{Z}, \tilde{\mathbf{z}} \neq \mathbf{z}^*, \tilde{\mathbf{z}} \in \mathcal{C},$$

then the ambiguity degree $\gamma = 1$. In other words, if there is an incorrect grounding $\tilde{\mathbf{z}}$ that always co-occurs with the correct one, then it is not possible to learn the correct classifier $p(\mathbf{z}|\mathbf{x})$ because it is not possible to determine which grounding is correct for $\mathbf{x}$. Therefore, this problem maps directly to the *small ambiguity degree condition*.

**MCMC Algorithm**

---

**Algorithm 1** MCMC for Grounding Planners

---

**Function** Step($\mathbf{z}$, $\nu$, $\mathcal{C}$)
 $\hat{\mathbf{z}} \leftarrow \text{Perturb}(\mathbf{z})$ ;            // Modify z randomly.
 $\boldsymbol{\zeta} \leftarrow \text{RandMask}(\hat{\mathbf{z}}, \nu)$ ;          // Mask perturbed z.
 $(\text{sat}, \mathbf{z}') \leftarrow \text{SAT}(\boldsymbol{\zeta}, \mathcal{C})$ ;         // Complete $\boldsymbol{\zeta}$.
 **if** *sat* **then**
  |  **return** $\mathbf{z}'$
 **else**
  |  **return** None
 **end**
**Function** RunChain($p_\theta, q, \mathbf{x}, n, \mathcal{C}$)
 $i \leftarrow 0$   $\mathbf{z} \sim p_\theta(\mathbf{z}|\mathbf{x})$ ;        // Initialization.
 $\mathbf{z} \leftarrow \text{Project}(\mathbf{z}, \mathcal{C})$ ;          // Fix z if invalid.
 **while** $i < n$ **do**
  $\nu \leftarrow 0.1$
  **while** $\nu \leq 1$ **do**
   $\mathbf{z}' \leftarrow \text{Step}(\mathbf{z}, \nu, \mathcal{C})$ ;       // Next sample.
   **if** $\mathbf{z}' = None$ **then**
    |  $\nu \leftarrow \nu + 0.1$ ;        // Mask more.
   **else**
    |  **break**
   **end**
  **end**
  $r \leftarrow \frac{q(\mathbf{z}'|\mathbf{x}, \mathcal{C})}{q(\mathbf{z}|\mathbf{x}, \mathcal{C})}$ ;        // Acceptance ratio.
  $\alpha \sim U(0, 1)$
  **if** $\alpha \leq r$ **then**
   |  $\mathbf{z} \leftarrow \mathbf{z}'$
  **end**
  $i \leftarrow i + 1$
 **end**

---

**Improved training stability of A2G**

Figure 2 (a) illustrates example training curves for the two methods, A2G and SoftenSG. The run with the KL objective alone exhibits instability, as just after epoch 600, the performance of the model collapses for a few epochs. While it does start to recover some time later, it is difficult to predict when and if that may occur and as a result standard convergence checks such as early stopping may terminate training. Figure 2 (b) shows the confusion matrix over the validation set at epoch 630, just after the performance drop.

**Additional training details**

In this section we provide additional details for model training in our experiments. We run all experiments on a server with 4 NVIDIA RTX 2080Ti GPUs, and an Intel(R) Core(TM) i9-9940X processor with 130 GB of memory. We use exponential decay for the $\gamma$ hyperparameter with an inital value of 2.

**Hand written formulas experiment:** As in the original SoftenSG work, we perform training in two stages in this experiment. We anneal $\gamma$ in *Stage 1* and set it to 0 in *Stage 2*, where we sample from $q$ by rejection sampling on $p_\theta$. We apply the score-matching term in both stages. We use the MCMC sampler with fixed projections for both methods and use the same hyperparameters as reported in the original experiments. Table 4 shows the per-stage breakdown of performance.

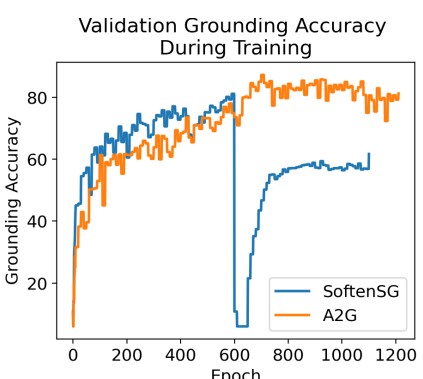
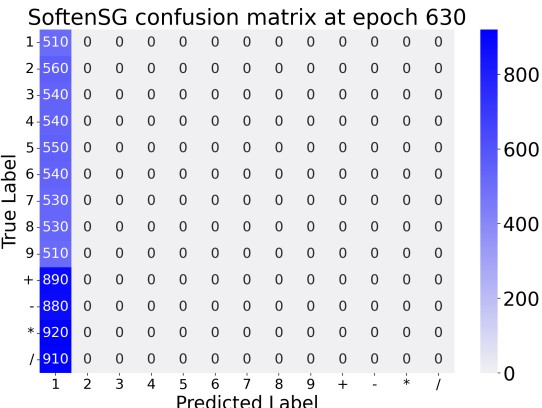

Figure 2: Example of training inconsistency using for training grounding models in the hand-written formula dataset without SM. Due to poor approximations, it is possible for the model during training to place low probability mass on valid groundings and get trapped due to uninformative gradients. (left) shows the validation accuracy for SoftenSG (blue) in comparison to A2G (orange) that uses the score-matching objective. (right) Shows the confusion matrix on the validation set for epoch 630 for SoftenSG. This indicates that at this step in the optimization, the model makes predictions inconsistent with the observations, likely placing low mass within the constraints of a given observation. As a result, it gets trapped for a few epochs, taking time to recover and leading to poor training.

| Method | Grounding Accuracy | Result Accuracy |
|---|---|---|
| SSL | $67.01 \pm 2.62$ | $6.29 \pm 1.49$ |
| SoftenSG - Stage 1 | $82.09 \pm 3.03$ | $25.43 \pm 6.58$ |
| SoftenSG - Stage 1+2 | $57.24 \pm 41.35$ | $44.55 \pm 43.89$ |
| A2G - Stage 1 | $89.00 \pm 1.90$ | $44.35 \pm 6.18$ |
| A2G - Stage 1+2 | $\mathbf{98.59 \pm 0.16}$ | $\mathbf{90.42 \pm 1.05}$ |

Table 4: Comparison of methods on the handwritten formula dataset, averaging over 8 runs for each method and reporting average and standard deviation grounding and result accuracy. The addition of the score-matching objective stabilizes training and results in better performance on average with high consistency.

For all methods we use the LeNet-5 architecture with batch size 64 and 10 steps of MCMC sampling. We use the same hyperparameters as Li et al. (2024) for SSL and the KL-only method. For G2A, the optimizer we use SGD learning rate 0.01 for Stage 1 and Adam with learning rate 1e-4 in stage 2. We set the $\alpha$ hyperparameter to 0.01 to control the loss balance. In all methods we use exponential annealing for $\gamma$ with initial value of 2. During stage 2 training, we set $\omega$ to infinity and use the same samples as KL, decreasing the probability of illegal samples and increasing the probability of legal ones.

**Maze experiment:** We use batch size 64 and 10 steps of MCMC sampling. For the optimizer we use Adam with learning rate 0.01. We use exponential annealing for $\gamma$ with initial value of 2. For G2A, we set the $\alpha$ hyperparameter to 0.01 to control the loss balance and make use of gradient clipping with norm of 1.

**Blocksworld experiment:** We use batch size 64 and 5 steps of MCMC sampling to control running time. For the optimizer we use Adam with learning rate 0.01. We use exponential annealing for $\gamma$ with initial value of 2. For G2A, we set the $\alpha$ hyperparameter to 0.001 to control the loss balance and make use of gradient clipping with norm of 1.

### Additional Analysis of Task Completion Performance

In this section, we provide additional quantitative results for the performance of the planning agent under various conditions. The analysis focuses on the agent's performance in the maze environment using the trained grounding models, as well as comparisons in the blocksworld domain.

#### Maze Environment

We begin by presenting the task completion performance of the agent in the maze environment, using the grounding models trained as described in the main paper (as seen in Table 5):

| Method | Task Success % |
|---|---|
| SoftenSG | $12.00 \pm 0.70$ |
| A2G | $14.90 \pm 3.21$ |
| Supervised | $20.40 \pm 4.80$ |

Table 5: Task success rates in the maze environment for different grounding methods.

For all methods, including the supervised one, the task completion performance is limited. This is due to the sensitivity of the planning process to errors in the grounding component, where even minor inaccuracies can result in invalid maze configurations (e.g., no cell being designated as the goal position). To address this, we employ a decoding procedure that refines the validity of the predicted outputs—a common approach in multi-label classification problems.

In this procedure, groundings are decoded by first selecting the agent's position based on the corresponding class logits for each maze cell. Next, the goal position is identified. For the remaining cells, each is classified as either clear or a wall based on the associated class probabilities. The application of this decoding strategy results in a marked improvement in task success rates (as seen in Table 6):

The decoding process notably enhances grounding performance, leading to improved task success rates. There are various methods to decode groundings from a trained model or integrate decoding within the training process. Since A2G does not impose constraints on the grounding model's parameterization, existing standard procedures can be applied, and we leave further exploration of these methods to future work.

| Method | Task Success % |
|---|---|
| SoftenSG | $68.90 \pm 2.98$ |
| A2G | $76.60 \pm 6.53$ |
| Supervised | $89.10 \pm 1.59$ |

Table 6: Task success rates in the maze environment after applying a decoding procedure to refine groundings.

Maze Per-Cell

As mentioned in Section 4.2 of the main paper, the challenge in the maze environment stems from the large output space. We intentionally chose to apply A2G to this challenging problem to highlight its performance improvements over previous approaches. A more standard approach would involve learning a "per-cell" classifier that independently predicts the state of each maze cell (i.e., clear, wall, agent, goal). This problem is significantly simpler, and both A2G and the baseline methods can achieve 100% grounding accuracy. Below, we show the results of this experiment, emphasizing the improved data efficiency of A2G (as seen in Table 7):

| Method | Task Success % | # of Interactions |
|---|---|---|
| SoftenSG | $100 \pm 0.00$ | $(2.84 \pm 0.31) \times 10^4$ |
| A2G | $100 \pm 0.00$ | $(1.44 \pm 0.20) \times 10^4$ |

Table 7: Task success rates and number of interactions required to achieve 100% accuracy in the simpler per-cell classification task in the maze environment.

Both methods achieve 100% task success eventually, as they both achieve 100% grounding accuracy. However, A2G requires significantly fewer environment interactions, demonstrating improved data efficiency.

Blocksworld Domain

Finally, we present the task performance of A2G without decoding in the blocksworld domain, including results for the larger blocksworld dataset (as seen in Tables 8 and 9):

| Method | Task Success % |
|---|---|
| SoftenSG | $0.00 \pm 0.0$ |
| A2G | $19.60 \pm 3.01$ |
| Supervised | $94.83 \pm 0.51$ |

Table 8: Task success rates in the Blocksworld dataset without decoding.

In both datasets, A2G significantly outperforms the baseline but falls short of the supervised benchmark. Similar to the maze environment, small errors in grounding can result in invalid predictions (e.g., two blocks being placed on the same block), preventing the

| Method | Task Success % |
|---|---|
| SoftenSG | $0.00 \pm 0.0$ |
| A2G | $67.18 \pm 5.32$ |
| Supervised | $100 \pm 0.00$ |

Table 9: Task success rates in the BlocksworldLarge dataset without decoding.

agent from forming a valid plan. Consequently, the baseline does not achieve sufficient accuracy to solve the task. A decoding procedure that respects such constraints could further improve the performance of both methods, and we consider this a direction for future work.

**Prompting
and settings for multimodal models.**

For grounding using multimodal models we iterated on different prompts and output strategies using the validation set of each dataset. In doing so, we decided to use structured output decoding through OpenAI's structured output API and the corresponding Ollama API to ensure controlled generation of symbols for the grounding task. We use the following prompts:
Maze domain:

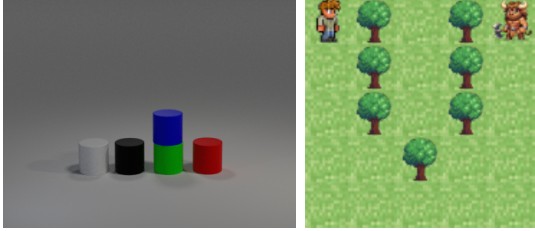

Figure 3: Example images from the photorealistic-blocksworld dataset (left) and maze environment (right).

> Describe the given image as a 5x5 maze. First find the player position. Then the robot position, corresponding to the goal. Then the trees and the empty positions. Then generate the maze. Make sure the maze you write matches the maze in the picture. First think in steps. Then provide your answer as a json, filling each row with the corresponding object.

Blocksworld domain:

> Describe this image in terms of On relationships between blocks. Always think in detail in steps, reasoning about the blocks. For each block, decide what is below it. Then write out the grid according to your reasoning as a json.

The structured output models are available alongside our training code.

