# OpenReview forum: "Act-to-Ground: A Framework for Symbol Grounding in Planning Domains"
_nesyconf.org/NeSy/2025/Conference — NeSy 2025 Poster_

### Official Review · Reviewer_CaWJ · 2025-04-03
**Intresting paper, some gaps in the related work**

**Rating:** 7
**Confidence:** 4

**Review:**

The paper proposes a system for neurosymbolic grounding of planning domains in subsymbolic observations. The approach relies on a SAT planner that weakly supervises the grounding process, leveraging the available knowledge. Additionally, it introduces an MCMC sampler for generating weak labels to guide the grounding process and a score-matching objective to improve grounding performance. The authors evaluate two training modes for their framework—one interactive and one offline—and demonstrate that their approach outperforms the baselines.

The paper tackles an important topic in AI and planning: symbolic grounding. It does so with a novel algorithm, which is presented in a fairly clear and structured manner.

However, the paper should discuss prior works that perform semi-supervised grounding using logical knowledge and interaction with the environment. For instance, [1] integrates such an approach into Reinforcement Learning problems.

Regarding the learnability condition, there is a rich body of literature on the issue of underdetermination in weakly supervised grounding from logical models, which the authors do not reference. Key related concepts include groundability [2] and reasoning shortcuts [3] [1]. The proposed learnability condition appears to be an application of these ideas specifically to planning. The authors should explicitly connect their work to this existing literature.

Minor issues:
- Some citations refer to the arXiv versions of papers instead of their published versions (e.g., Daniele et al. 2023 and Garnelo et al. 2016). If the papers are officially published, they should be cited with the correct venue.
- On page 9, the reference to Table 8 likely refers to Table 2 instead.

That said, these weaknesses can be addressed in the camera-ready version. Overall, I believe the approach is solid and the paper well-written, and it deserves to be published at this venue.

References:

[1] Umili et al. ECAI 2024, “Neural Reward Machines”

[2] Umili et al. KR 2024, “Grounding LTLf specifications in image sequences”

[3] Marconato et al. NeurIPS 2023, “Not all neuro-symbolic concepts are created equal: Analysis and mitigation of reasoning shortcuts”

**Anonymity:**

Remain anonymous

---

### Official Review · Reviewer_cGng · 2025-04-05
**A Framework for Symbol Grounding in Planning Domains" (Act-to-Ground) proposes a new method using satisfiability-based planning for weak supervision to train symbol grounding models in planning domains, showing significant improvements over baselines and introducing a learnability condition.**

**Rating:** 7
**Confidence:** 3

**Review:**

The paper presents a theoretically sound framework, Act-to-Ground (A2G), for addressing the Symbol Grounding Problem (SGP) in neurosymbolic planning. The approach of using satisfiability-based planning (SAT-planning) to provide weak supervision to a grounding model is novel and well-motivated. The introduction of an MCMC sampler for weak label sampling and the incorporation of a score-matching objective to improve training stability and performance are significant contributions. The paper also formally (and informally) presents a learnability condition for neurosymbolic symbol grounding. The empirical evaluation across three distinct experimental settings (maze navigation, Photorealistic Blocksworld, and handwritten formulas) provides strong evidence for the effectiveness of the proposed A2G framework and the score-matching objective, with comparisons against relevant baselines and supervised benchmarks.

Clarity:
The paper is well-written and logically organized. The introduction clearly articulates the SGP and the motivation for the A2G framework. The "Related Work" section thoroughly discusses relevant prior research. The "Method" section provides a detailed explanation of the A2G framework, including the use of SAT-planning for weak supervision, the MCMC sampler, the score-matching objective, and the learnability condition. Figures and Algorithm 1 aid in understanding the architecture and the MCMC sampling process. The experimental setup and results are clearly presented with relevant metrics.

Originality:
The paper presents several original contributions to the field of neurosymbolic learning and symbol grounding:

• The core idea of using satisfiability-based planning to generate weak supervision for training grounding models is a novel approach to address the data scarcity problem.

• The proposed MCMC sampler with randomized projections offers a practical method for sampling weak labels for grounding planning problems, especially in high-dimensional state spaces.

• The introduction and demonstration of the effectiveness of a score-matching objective in improving the training consistency and performance of neurosymbolic grounding models is a significant contribution.

• The formulation of a learnability condition that connects the learnability of neurosymbolic symbol grounding for planning agents to the consistency of supervision provides valuable theoretical insight.

• The evaluation of the A2G framework in two distinct learning settings (learning by interaction and learning from demonstrations) within planning domains showcases its versatility.

Significance:
The paper tackles the fundamental Symbol Grounding Problem, a major challenge in the development of transparent and interpretable neurosymbolic systems, particularly for planning. The A2G framework offers a promising approach to train grounding models without the need for extensive labeled data by leveraging the inherent knowledge within symbolic planning domains and weak supervision from environment interaction or demonstrations. This has the potential to significantly broaden the applicability of neurosymbolic planners to real-world tasks, such as robotics. The improved training stability achieved through the score-matching objective addresses a critical issue in weakly supervised learning in high-dimensional spaces. The theoretical analysis of learnability provides a deeper understanding of the conditions under which accurate grounding models can be learned with weak supervision.

Pros:

• Presents a novel and theoretically grounded framework (A2G) for symbol grounding in planning domains using weak supervision.

• Effectively leverages satisfiability-based planning to generate weak supervision, reducing the reliance on labeled data.

• Introduces an innovative MCMC sampler for efficient weak label sampling.

• Demonstrates that the score-matching objective significantly improves training stability and grounding accuracy.

• Provides a learnability condition offering theoretical insights into the problem.

• Evaluates the framework rigorously across diverse experimental settings and problem domains, showing significant improvements over baselines.

• Addresses a critical challenge (Symbol Grounding Problem) for the broader adoption of neurosymbolic AI.
• Has the potential to expand the applicability of symbolic planners to tasks where obtaining fully labeled data is infeasible.

Cons:

• Task completion performance in complex planning environments (maze, Blocksworld) is still sensitive to grounding errors, indicating that even small inaccuracies can hinder successful planning. While the paper explores decoding strategies, further work is needed in this area.

• The learnability condition suggests that weak supervision alone might not be sufficient in all scenarios, and supplementary techniques (e.g., unsupervised learning) might be necessary.

• The current implementation of A2G does not fully utilize sequential information from trajectories in the learning by interaction setting, which could be a limitation in partially observable environments.

• The computational cost associated with SAT-solving within the MCMC sampling process could be a factor in scaling the framework to very large and complex planning domains.

**Anonymity:**

Remain anonymous

---

### Official Review · Reviewer_uBMC · 2025-04-05
**Review for A2G**

**Rating:** 5
**Confidence:** 4

**Review:**

This work presents a method for symbol grounding using weak supervision.
The method is applied in two different contexts, namely learning by interaction and learning by demonstration.
Each of those being respectively evaluated with mazes and blocksworld.
The problem is a relevant issue for NeSy AI community and is a very interesting problem.
The authors demonstrate higher performance on two tasks than the recent work SoftenSG.
Moreover, the authors compare against current closed and open foundation models.
The problem is formulated under probabilistic inference terms, and an approximation to the true posterior is computed using a combination of variational inference and MCMC.
The text does provide a good evaluation of the results, specially considering multiple models and experiments discerning between fully supervised and weakly supervised.

Some of the elements that the reviewer considers it could increase the general quality of the paper:
* Section 3.2 is hard to follow without introducing the MCMC sampling scheme.
* Is rather unclear why both MCMC and VI are needed.
* Elaborating beforehand about what elements explicitly constitute the constrained set C would be beneficial.
* Elaborating in more detail about the explicit contents of the planning domain $\Sigma$, the constraints $C$, the model $p_{\theta}$, and the parameters $\theta$ would be highly beneficial for the reader.
* Elaborating how the SAT solver is applied would be very interesting for readers not familiar in this domain.
* Figure 1 could showcase the explicit values of each arrow, and not only the weak labels.
* Learning by interaction and from demonstration are only explicitly shown in the middle of the paper and are not foretold in the introduction. It could be beneficial to the reader to know beforehand that types of problems are being considered
* The text uses a form that might not be suitable for scientific discussion "our hope is".
* On a first read is unclear why Theorem 1 is introduced in this section. A brief explanation beforehand would be suitable for the reader.
* Moreover, an informal Theorem definition might not be suitable. An alternative would be informally talk about the assumptions and the implications, and then proceed to formally define the theorem.

Open questions that the reviewer has:
* What type of domain information is provided to the planner in both tasks?
* The score-matching weight alpha most likely is a relevant parameter for the final performance measure of both tasks. Was a hyper-parameter tuning methodology applied? Is task's performance subject to large discrepancies if this parameter is modified?
* What model is used to represent the density of $p_{\theta}$, and what constitutes the parameters $\theta$?
* Both VI and MCMCM are algorithms used for probabilistic inference. Usually either one of those is applied to approximate posterior distributions. Why were both algorithms required for this task?
* Why would MCMC alone not be applicable?
* Are the guarantees of MCMC convergence continue hold when the masking procedure is done inside the chain?
* What types of limitations exists within A2G when larger problems are considered? Explicitly when the constraints C need to be continually checked.

**Anonymity:**

Remain anonymous